# Bioaccumulation of PCBs and PBDEs in Fish from a Tropical Lake Chapala, Mexico

**DOI:** 10.3390/toxics9100241

**Published:** 2021-09-29

**Authors:** Ernesto Oregel-Zamudio, Dioselina Alvarez-Bernal, Marina Olivia Franco-Hernandez, Hector Rene Buelna-Osben, Miguel Mora

**Affiliations:** 1Instituto Politécnico Nacional, Unidad Profesional Interdisciplinaria de Biotecnología-IPN (UPIBI), Av. Acueducto, Barrio la Laguna Ticoman, Ciudad de México 07340, Mexico; mofrancoh@hotmail.com; 2Instituto Politécnico Nacional, Centro Interdisciplinario de Investigación para el Desarrollo Integral Regional, Unidad Michoacán, Justo Sierra 28, Col. Centro, Jiquilpan 59510, Mexico; dalvarezb@ipn.mx (D.A.-B.); hrbuelna@hotmail.com (H.R.B.-O.); 3Department of Wildlife and Fisheries Sciences, Texas A&M University, 454 Throckmorton St, College Station, TX 77840, USA; mmora@tamu.edu

**Keywords:** POPs, sediments, water, *Chirostoma* spp., *Cyprinus* *carpio*, *Oreochromis* *aureus*

## Abstract

Lake Chapala is the largest natural freshwater reservoir in Mexico and the third largest lake in Latin America. Lakes are often considered the final deposit of polluting materials; they can be concentrated in the organisms that inhabit them, the water, and the sediments. The PCBs and PBDEs are environmental pollutants highly studied for their known carcinogenic and mutagenic effects. PCB and PBDE bioaccumulation levels were determined in *Chirostoma* spp., *Cyprinus carpio*, and *Oreochromis aureus.* In addition, we monitored the concentrations of PCBs and PBDEs in sediment and water from Lake Chapala were monitored. Samples were collected during two periods, in October 2018 and May 2019. The samples were analyzed by gas chromatography coupled with mass spectrometry. Two bioaccumulation factors were determined in fish, one in relation to the concentration of PCBs and PBDEs in sediments and the other in relation to the concentration of PCBs and PBDEs in water. The PCB levels were 0.55–3.29 ng/g dry weight (dw) in sediments, 1.43–2.98 ng/mL in water, 0.30–5.31 ng/g dw in *Chirostoma* spp., 1.06–6.07 ng/g dw in *Cyprinus carpio*, and 0.55–7.20 ng/g dw in *Oreochromis aureus*. The levels of PBDEs were 0.17–0.35 ng/g dw in sediments, 0.13–0.32 ng/mL in water, 0.01–0.23 ng/g dw in *Chirostoma* spp., 0–0.31 ng/g dw in *Cyprinus carpio*, and 0.1–0.22 ng/g dw in *Oreochromis aureus*. This study provides information for a better understanding of the movement, global distribution, and bioaccumulation of PCBs and PBDEs. The results show that the fish, water, and sediments of Lake Chapala are potential risks to the biota and the local human population.

## 1. Introduction

Polychlorinated biphenyl compounds (PCBs) and polybrominated diphenyl ethers (PBDEs) are persistent organic pollutants (POPs) that are seemingly ubiquitous due to their volatility, multiplicity of sources, and transport mechanisms [1], and in the past 30 years, we have seen a significant increase in the number of places where they have been detected [2]. Contamination of aquatic ecosystems with PCBs and PBDEs is growing at an alarming rate and has become an important problem worldwide [3,4]. Despite vast evidence of the serious damage and deterioration that PCBs and PBDEs cause in terrestrial and aquatic ecosystems around the world, they have been studied very little in Mexico. Although the production and use of PCBs and PBDEs are now prohibited in many parts of the world, they continue to be present in the environment from past use and manufacture, either released by spills or accidental leaks or from (often illegal) disposal in inappropriate waste sites and improvised dumps [5,6]. Additionally, they were released by anthropogenic actions, such as the inappropriate disposal of old electronic equipment, incineration of materials that contain them, vehicular emissions, and pluvial runoff from local dumps [7]. In addition to being toxic and persistent, PCBs and PBDEs bioaccumulate, which increases the need for consistent monitoring efforts.

Lake Chapala is the largest lake in Mexico, the third largest in Latin America, and the second highest in America, surpassed in altitude only by Lake Titicaca. Lake Chapala provides important economic, recreational, and ecosystem services, with nearly 10% of the population of western Mexico using resources from this lake [6,8]. Lake Chapala has been exposed to different sources and levels of pollution from agricultural runoff and industrial and municipal wastewater from many communities, mainly along the Lerma River [9]. Lake Chapala is cataloged as one of the most threatened natural resources in the western hemisphere by the United Nations Environmental Program; at the same time, since 2009 it has been considered one of the most important wetlands in Mexico (RAMSAR Site no. 1973) [10].

Lake Chapala, in addition to being a valuable source of fresh water, is also an important commercial fishing zone. In recent years, the consumption of fish in the region has increased due to its nutritional benefits to consumers as a source of proteins, essential minerals, vitamins, and unsaturated fatty acids. Organizations, such as the American Heart Association, even recommend consuming fish once or twice a day to reach a healthy daily intake of omega-3 fatty acids [11]. Notwithstanding, some publications have shown that consuming fish from this region may be dangerous. Several reports describe an accumulation of mercury (Hg) in fish from Lake Chapala [9,12], and there is evidence that Lake Chapala sediments, water, and fish have higher than ideal concentrations of metals (Al, Ba, Cu, Mn, Hg, Sr, V, and Zn) [13]. Although the concentrations of metals in Lake Chapala were below levels that would on their own generate concern about negative effects, they are coupled with evidence of possible contamination with POPs [6,14]. Considering that aquatic ecosystems are the endpoint of nearly all chemical substances, the possibility of accumulation of PCBs and PBDEs is a legitimate concern in this ecosystem.

Although existing studies of pollution in Lake Chapala have not focused broadly on POPs, PCB and PBDE concentrations have been quantified in lake sediments and documented increases in the concentrations of these contaminants since the 1990s despite the prohibitions on their use. There is evidence of moderate to high concentrations of POPs in the lake, which suggest that the concentrations in sediments could be harmful to humans through fish consumption [6]. In addition to their relevance, given the potential harm to humans from consuming them, the fish of Lake Chapala could be reliable indicators of bioaccumulation of PCBs and PBDEs. We chose three fish species of ecological, economic, and commercial importance to monitor PCBs and PBDE concentrations: *Chirostoma* spp. is a pelagic predator and a characteristic small organism of Lake Chapala; *Cyprinus carpio* is located at the highest levels of the trophic web of Lake Chapala and has benthic habits; and *Oreochromis aureus* is the most abundant and most consumed species, as well as a potentially invasive species [15].

To date, there are no studies that quantify the levels of PCBs and PBDEs in both the water and fish, or that calculate the degree of bioaccumulation of these POPs in aquatic organisms of Lake Chapala. Considering, on the one hand, the importance of Lake Chapala as a regional supply of fresh water and commercial fishing, and on the other hand, the evidence of contamination of its sediments with PCBs and PBDEs, the objective of this study was to determine the concentrations of PCBs and PBDEs in the water, sediments, and three fish species (*Chirostoma* spp., *Cyprinus carpio*, and *Oreochromis aureus*) of Lake Chapala and the degree of bioaccumulation in fish.

## 2. Materials and Methods

### 2.1. Study Area and Sampling

Lake Chapala is located at the coordinates 20°13′ N 103°03′ W, has an area of 114,659 ha, and has an altitude of 1510 masl. According to its biophysical characteristics, this lake is defined as neotectonic, mesotrophic, and tropical, with a catchment area of 52,000 km^2^ and average depth of 7 m. It belongs the most developed fluvial basin in Mexico, the Lerma–Santiago–Pacific basin, which has an approximate drainage area of 140,000 km^2^. Its main source of water input is the Lerma River (annual flow volume of 68.2 m^3^s^−1^), but it also receives 2.4 m^3^s^−1^ of wastewater directly. The natural outlet is the Grande de Santiago River, which flows directly into the Pacific Ocean. The climate is temperate sub-humid with rains in the summer (annual precipitation of 700–1200 mm, mean evaporation o f1700 mm), and mean temperatures range from 5 to 29 °C.

Fish, water, and sediment samples were collected in two local seasonal periods, October 2018 and May 2019, which corresponded to the rainy and dry seasons, respectively. There were 42 sampling stations for water and sediments (Figure 1). At each station, water and sediment samples were collected in duplicate. The water samples were collected at a depth of approximately 0.5 m. The sediment samples were collected from the bottom sur-face of the lake using an Ekman dredge. There were five fish sampling stations that corresponded to the local fishing places in the lake (Figure 1); in each station, four samples of *Chirostoma* spp., *Cyprinus carpio*, and *Oreochromis aureus* were collected. All samples were transported in coolers and kept in amber glass containers until their use.

### 2.2. Chemical Substances and Reagents

Dichloromethane (Sigma-Aldrich, St. Louis, MO, USA) was used as the extraction solvent. The samples were purified using calcium chloride, silica, and alumina (Sigma-Aldrich, St. Louis, MO, USA). For the identification and quantification of PCBs, a mixture of 39 authenticated standards was used, corresponding to the following congeners: 1, 2, 3, 4, 5, 7, 9, 16, 18, 19, 22, 25, 28, 44, 52, 56, 66, 67, 71, 74, 82, 87, 99, 110, 138, 146, 147, 153, 173, 174, 177, 179, 180, 187, 194, 195, 199, 203, and 206 (AccuStandard, New Haven, CT, USA). For the identification and quantification of PBDEs, a mixture of 39 authenticated standards was used corresponding to the following congeners: 1, 2, 3, 7, 8, 10, 11, 12, 13, 15, 17, 21, 28, 30, 32, 33, 35, 37, 47, 49, 66, 71, 75, 77, 85, 99, 100, 116, 118, 119, 126, 137, 153, 154, 155, 166, 182, 183, and 190 (AccuStandard, New Haven, CT, USA). As an internal standard, decachlorobiphenyl (Sigma-Aldrich, St. Louis, MO, USA) was used. A mixture of PCBs (180, 138, 101, 153, 118, and 28) and PBDEs (47, 99, 100, and 153) congeners was used as recovery standards (Sigma-Aldrich, St. Louis, MO, USA).

### 2.3. Physicochemical Analyses

The physicochemical parameters of the water were measured in situ using a portable multiparameter meter (Pro Plus, YSI, Yellow Springs, OH, USA). The size of each fish was measured in terms of its weight, using an analytical balance, and length, width, and tail length with an electronic caliper. The sampled fish were then ground completely and dried in a lyophilizer [17]. To determine the total fat and total protein of the lyophilized fish samples, we used a near-infrared apparatus [18]. The total fat values were validated using a Soxhlet apparatus, using ether as the solvent [19]. The protein values were validated using the protein content estimated from the total nitrogen content (N*6.25) obtained by the Kjeldahl method [20].

### 2.4. PCBs and PBDEs Analysis

The sediments were lyophilized, and then ground to powder using a porcelain mortar. The fish samples were ground completely, then lyophilized, and ground to powder. All of the samples were extracted with dichloromethane by the reflux extraction technique using a Soxhlet apparatus. The sample extracts were then purified by chromatography in a column containing a 2:1 mixture of silica to alumina. Then, the purified extracts were concentrated in a Rotavapor (RII, Büchi, Flawil, St. Gallen Switzerland) and transferred to amber glass vials for analysis.

A gas chromatograph (Clarus 680, Perkin-Elmer Inc., Waltham MA, USA), coupled with a mass spectrometer (Clarus SQ8T, Perkin-Elmer Inc., Waltham, MA, USA) was used in selective ion monitoring mode. The samples were injected separately into two specific capillary columns: ZB-MultiResidue-1 (Zebron™ ZB-MultiResidue™-1, GC Cap. Column 30 m × 0.32 mm × 0.50 µm) and ZB-MultiResidue-2 (Zebron™ ZB-MultiResidue™-2, GC Cap. Column 30 m × 0.32 mm × 0.25 µm). The run conditions for the gas chromatograph were: injection by autosampler in splitless mode, injector temperature 250 °C, helium carrier gas at a constant flow rate of 1 mL/min; the oven temperature was raised from 120 °C for 0.5 min to 210 °C @ 30 °C/min to 300 °C @ 6 °C/min for 5 min. The conditions of the mass spectrometer were: transfer line temperature 250 °C, ionization source temperature 250 °C, ionization voltage 70 eV; scan time was 0.3 s with a scanning delay of 0.5 s and a solvent delay of 5 min. The data were recorded and processed using Turbo Mass software (version 6.1.0.1963, Perkin-Elmer Inc., Waltham, MA, USA).

Problem analytes by verifying the respective retention times of the quantification ions in the authenticated standards. Moreover, the confirmation ions were compared and verified, comparing the mass spectra of each analyte to the NIST mass spectral library (NIST/EPA/NIH Mass Spectral Library version 2.2. 2014). To quantify the analytes, standard curves were constructed by injecting each of the authenticated standards at five different concentrations. The concentrations of the problem analytes were calculated using the calibration equations obtained from these standard curves based on the relative concentrations and areas, using the recovery standards and internal standards as references. The area and concentrations of the recovery and internal standards were obtained routinely from independent samples in each lot. A blank procedure was performed in duplicate, and the replicated sample had variations within acceptable limits (<±25%). The process of identification and quantification for each of the cases was conducted using chromatographic and spectral data obtained from the two individually used columns.

The recovery percentages of the surrogate standards for PCBs varied as follows: 180: 55–112%, 138: 44–108%, 101: 77–115%, 49: 88–95%, 153: 63–103%, 118: 48–91%, and 28: 64–105%. The recovery percentages of surrogate standards for PBDEs varied as follows: 47: 84–95%, 99: 73–91%, 100: 75–89%, and 153: 78–97%. The limits of detection (LOD) of each PCB and PBDE congeners were calculated as the lowest mass present in the extracted sample of each analyte, which provided an instrumental response equal to the signal observed with the lowest calibration solution. The LODs ranged from 0.01 to 0.09 ng/g for PCBs, 0.01 to 0.05 ng/g for PBDEs.

### 2.5. Bioaccumulation Factor

The bioaccumulation factors of the analytes in fish were determined relative to the dissolved concentration of the analytes in the sediment (sediment-based bioaccumulation factor; SBF), and in the water samples (water-based bioaccumulation factor; WBF). These were taken as the base for concentrations of each PCB and PBDE congener individually in each sample from each fish species, water, or sediment, for 2018 and 2019. The SBF and WBF were calculated following Aboul Ezz and Abdel-Razek [21], Authman and Abbas [22], and Badii et al. [23], by dividing the concentration of each of the chemical compounds in the tissue by the concentration of the same compound in the sediment (for SBF) or water (for WBF).

### 2.6. Univariate and Multivariate Statistical Analysis

The graphs and multivariate analyses were made in the software RStudio (Version 1.3.1093, 2009–2020 RStudio) using R version 4.0.2 and the packages Plotly and MetaboAnalystR 4.0. The dataset of the PCB and PBDE concentrations were constructed in Excel and exported to RStudio. For the multivariate analysis, the values of the analytes reported as not detected (ND) were replaced with 1/5 of the positive minimum value of each corresponding variable. The normal distribution of the sample dataset was verified to make the different matrices and seasons comparable. Interactive scatterplots with bubble markers were made to represent the concentrations of the different analytes, the interaction between matrices, and the seasons. Heat maps with hierarchical groupings of the analyzed samples were elaborated by season and by the congeneric analytes of PCBs and PBDEs respectively. For the groupings, an analysis of variance was performed (ANOVA) and a *t*-test, a measure of Euclidean distance, and a Ward grouping algorithm were used.

## 3. Results

### 3.1. Physicochemical Parameters

Although the monitoring of the physicochemical parameters of lakes is a routine process in many parts of the world, in Mexico it is sporadic, mainly because of economic conditions. Lake Chapala, being in the northern tropics, is subjected to strong cycles of the rainy-dry seasons. The approximate annual change can typically be up to 3 m between stations. About half of the lake’s water output is pumped into the surrounding urbanized areas. Due to the relatively small volume of the lake and its large surface area, the remaining output is evaporation. Therefore, important changes can occur between seasons and there could be changes in the transport and bioaccumulation patterns of PCBs and PBDEs.

The sediment pH ranged from 7.45 to 7.30, the electrical conductivity from 1.01 to 1.24 dS/m, total carbon from 2.25 to 2.59%, nitrogen from 0.15 to 0.21%, and organic matter 3.20–3.54%. There were no significant differences between 2018 and 2019 in the sediment physicochemical parameters. Similarly, in the water samples we observed, no significant differences between 2018 and 2019 for temperature (23–24 °C), dissolved oxygen (5.12–5.26 mg/mL), electrical conductivity (767–869 µS/cm), or total soluble solids (519–581 mg/L). However, the salinity (0.000388–0.000438 ng/L) and nitrate (6.57–13.54 mg/L) values did differ significantly between the two years. The percent fat, protein, weight, and size of *Chirostoma* spp., *C. carpio*, and *O. aureus* are presented in Table 1. The three species differed in these parameters; *C. carpio* had the highest values, followed by *O. aureus*, then *Chirostoma* spp. (Table 1).

### 3.2. Polychlorinated Biphenyl Compounds (PCBs)

PCBs accumulated in fresh water and sediments originate mainly from their industrial application in paints, insulation additives, and dielectric fluids. Generally, between 50 and 100 congeners are released into the environment during the destruction and dis-mantling of electrical equipment. The congeneric PCBs show a wide range of known physicochemical properties that determine their transport pathways or bioaccumulation in organisms. PCBs are considered indicator pollutants for trend and risk assessments.

Table 2 shows the results of the PCBs analyses in samples of sediment, water, *Chirostoma* spp., *C. carpio*, and *O. aureus* collected in October 2018 and May 2019. In 2018, average concentrations were found, of 2.47 ± 0.25 ng/g dry weight (dw) in sediments and 2.29 ± 0.31 ng/mL in water samples. The concentrations in *Chirostoma* spp., *C. carpio*, and *O. aureus* were 1.32 ± 0.35, 1.72 ± 0.33, and 1.58 ± 0.31 ng/g dw, respectively. In addition, the average concentrations of PCBs in *Chirostoma* spp., *C. carpio*, and *O. aureus* were higher in 2019, approximately triple the average values from 2018 (4.22 ± 0.59, 4.66 ± 0.85 and 4.85 ± 0.93 ng/g dw).

In 2018, it was observed that the SBF values in *Chirostoma* spp. ranged from 0.13 to 0.83, in *C. carpio* from 0.43 to 1.05, and in *O. aureus* from 0.24 to 0.83. The highest levels were found in *C. carpio*. In general terms, the SBF values from 2019 were higher for all three fishes than in 2018: *Chirostoma* spp. (1.00–2.28), *C. carpio* (1.12–2.95), and *O. aureus* (1.16–2.84). The average values of SBF in 2018 were 0.54 ± 0.14, 0.71 ± 0.16, and 0.64 ± 0.12 for *Chirostoma* spp., *C. carpio*, and *O. aureus*, respectively. In 2019, the values obtained for SBF were 1.65 ± 0.32, 1.83 ± 0.47, and 1.89 ± 0.44 for *Chirostoma* spp., *C. carpio*, and *O. aureus*, respectively. The values obtained for WBF in 2018 were 0.58 ± 0.17 in *Chirostoma* spp., 0.76 ± 0.16 in *C. carpio*, and 0.70 ± 0.17 in *O. aureus*. For 2019, the averages obtained for WBF in the fish samples were significantly higher compared to 2018; that is, *Chirostoma* spp. 2.17 ± 2.38, *C. carpio* 2.38 ± 0.55, and *O. aureus* 2.48 ± 0.56. The WBF for samples of *Chirostoma* spp., *C. carpio*, and *O. aureus* were within the ranges of 0.14–0.81, 0.42–1.03, and 0.22–1.00, respectively, for the year 2018. For 2019, the WBF for *Chirostoma* spp., *C. carpio*, and *O. aureus* were within the ranges of 1.29–3.18, 1.36–3.79, and 0.98–3.63, respectively (Appendix A).

Hierarchical groupings were made among the SBF and WBF values of the three fish species in 2018 and 2019. Overall, there was a pattern of grouping by year; that is, there were differences between 2018 and 2019. There were no significant differences within 2018; however, in 2019 there were groupings based on SBF and WBF, respectively. In addition, generally, there were two large main groups. There was an evident grouping of SBF among fish samples from 2019, which were statistically different from the rest of the samples. A significant difference was observed in *O. aureus* with respect to *C. carpio* and *Chirostoma* spp., which did not have significant differences in SBF in 2019. A similar grouping pattern was observed for WBF in 2019, in which *O. aureus* also showed significant differences with respect to the other two species. No significant differences were observed between WBF and SBF in 2018; that is, in that year, the statistical groupings were mainly a function of the fish species and there were clear statistical differences among *O. aureus*, *Chirostoma* spp., and *C. carpio* (Figure 2).

### 3.3. Polybrominated Diphenyl Ethers (PBDEs)

PBDEs are a class of chemicals widely used as flame-retardants. This class of pollutants is also of global interest, because they have been found in the environment, in some populations of marine mammals and in humans. Although there is evidence of the presence of these substances in many different parts of the world, the consequences of exposure to PBDEs on human health are not well documented, because in all cases of contamination, human populations have been exposed to low levels of these substances. However, these substances, present in low concentrations in the environment and in organisms, are a risk.

Table 3 shows the values of average PBDE concentrations in Lake Chapala. The average PBDE concentrations were 0.25 ± 0.04 ng/g dw in sediments and 0.23 ± 0.03 ng/mL in water in 2018. The average concentrations of PBDEs were 0.26 ± 0.04 ng/g dw in sediments and 0.20 ± 0.03 ng/mL in water samples in 2019. There was an increase of approximately 5% in PBDE concentrations in sediment samples and a decrease of approximately 14% in water samples from 2018 to 2019. The average concentrations in samples of *Chirostoma* spp., *C. carpio*, and *O. aureus* for 2018 were 0.15 ± 0.03, 0.19 ± 0.05, and 0.16 ± 0.03 ng/g dw. The average concentrations in samples of *Chirostoma* spp., *C. carpio*, and *O. aureus* for 2019 were 0.06 ± 0.04, 0.08 ± 0.05, and 0.07 ± 0.05 ng/g. There was a decrease in PBDE concentrations of 60% in *Chirostoma* spp., 57% in *C. carpio*, and 56% in *O. aureus*, from 2018 to 2019.

The SBF values for PBDEs from 2018 were 0.31–1.18 for *Chirostoma* spp., 0.43–1.31 for *C. carpio*, and 0.34–1.09 for *O. aureus*. In 2019, the SBF values were in the ranges of 0.04–0.66, 0.01–0.65, and 0.02–0.72 for the samples of *Chirostoma* spp., *C. carpio*, and *O. aureus* respectively. The WBF for 2018 varied in *Chirostoma* spp. from 0.43 to 1.16, in *C. carpio* from 0.41 to 1.36, and in *O. aureus* from 0.37 to 1.07. In 2019, the values of WBF in *Chirostoma* spp., *C. carpio*, and *O. aureus* ranged from 0.04 to 0.89, 0.02 to 1.07, and 0.03 to 1.05, respectively. In the case of SBF for PBDEs in 2018, there were no significant differences among the three fish species (0.62 ± 0.19, 0.77 ± 0.20, and 0.68 ± 0.17 for *Chirostoma* spp., *C. carpio*, and *O. aureus*, respectively). However, in 2019, there were significant differences among the SBF values calculated for the three fish species; *C. carpio* (0.29 ± 0.19) had a higher value than *Chirostoma* spp. (0.23 ± 0.15) and *O. aureus* (0.25 ± 0.19). The average WBF values of for 2018 were 0.66 ± 0.17 for *Chirostoma* spp., 0.84 ± 0.24 for *C. carpio*, and 0.73 ± 0.16 for *O. aureus*. The average values for *Chirostoma* spp., *C. carpio*, and *O. aureus* were 0.33 ± 0.24, 0.42 ± 0.31, and 0.36 ± 0.30, respectively (Appendix A).

Figure 3 shows the heat map with hierarchical groupings as a function of SBF and WBF calculated from PBDE concentrations corresponding to *Chirostoma* spp., *C. carpio*, and *O. aureus*. There was a grouping of SBF corresponding to 2019; within this grouping, *O. aureus* was different from *Chirostoma* spp. and *C. carpio*, which did not show significant differences. A similar grouping pattern was observed for WBF 2019; there were no significant differences between *O. aureus* and *C. carpio*, but *Chirostoma* spp. was different in this parameter.

In general terms, relatively low values of bioaccumulation of PBDEs were observed in the study species, only some of which exceeded a value of 1. Figure 3 shows the most important analytes in terms of bioaccumulation, which in the case of SBF for 2018, PBDE 12 (1.18), and PBDE 21 (1.15) were the most important for *Chirostoma spp*. In the case of *C. carpio*, the most important PBDEs were PBDE 21 (1.06), PBDE 37 (1.13), PBDE 49 (1.02), PBDE 71 (1.31), and PBDE 118 (1.03). Observing *O. aureus* shows that PBDE 1(1.09) and PBDE 10 (1.00) were the most important in terms of bioaccumulation. In the same sense, the SBF values from 2019 were lower than those from 2018, and all of the values were below 1.00.

## 4. Discussion

### 4.1. Levels of PCBS and PBDEs in Lakes

Pollution is a topic of global interest; it has been shown that monitoring POPs in different parts of the world helps to understand their dynamics. Although it is well known that PCBs and PBDEs are transported long distances atmospherically, they are also known to bioaccumulate, and pose a risk to the local biota and populations. The levels of PCBs and PBDEs in this study were below the concentrations reported in the highly contaminated urban lakes in Mexico, such as Lake Mecoacán in Tabasco (PCBs: 6–372 ng/g dw, PBDEs: NA) [24] and Lake Chalco, in Mexico City (PCBs: 621 ng/g dw, PBDEs: NA). However, PCB and PBDE concentrations were higher in Lake Chapala than other urban and rural lakes in Mexico, such as Lake Tule (PCBs: 1.7–24.7 ng/g dw, PBDEs: 0.3–1.5 ng/g dw) and Lake Santa Elena (PCBs: 1.5–15.4 ng/g dw, PBDEs: 0.4–1.8 ng/g dw) [7] in Jalisco. By comparing the concentrations of PCBs in sediments, we found that, with water bodies elsewhere in the world, they were low compared to San Diego Bay, California (23–1387 ng/g dw) [25], the Pearl River Estuary in China (17.68–169.26 ng/g dw) [26], and coastal Bangladesh (32.17–199.4 ng/g dw) [27], and comparable to values found in the Yellow Sea (0.51–5.84 ng/g dw) [28] and the Hugli Estuary in India (0.18–2.33 ng/g dw) [29].

### 4.2. PCBs and PBDEs in Sediments

In surface sediments, we found PCB levels up to 3.29 ng/g dw and PBDE levels of up to 0.34 ng/g dw. These results are similar to another recent study reporting the concentrations of PCBs (0.3–27.1 ng/g dw) and PBDEs (0.2–2.5 ng/g dw) in Lake Chapala [6]. The sediments were composed mainly of silt (77–87%) and clay (8–19%) [3]. This composition indicates a mechanism of transport and distribution of PCBs and PBDEs in Lake Chapala, in addition to a form of protection and storage of the analytes within the particles of sediments with high organic matter [30]. However, the distribution and transport of the different PCBs and PBDEs congeners are directly related to the hydrodynamics of the lake system. In this study, the concentrations of PCBs and PBDEs were higher in sediments than in water, which could be due to the colloidal particles found in fine sediments, which have a higher absorption capacity for lipophilic compounds compared to small particles suspended in water [31].

The level of organic matter in the surface sediments indicates possible conservation of these persistent analytes in the lake, but the low oxygen concentration in most cases could influence the proportion of respective congeners between the matrices of the surface sediments and the water. Although we found the highest concentrations of PCBs and PBDEs in sediments in this study, it is likely that the rate of anaerobic biodegradation is highest in the sediment, since the sediments have strongly anoxic conditions with limited water exchange and considerable levels of eutrophication [32,33]. In addition, it should be considered that some lignolytic bacteria and fungi would find adequate substrate conditions for their development [34]. PCBs and PBDEs with a high degree of chlorination or bromination (hepta, octa, and deca) tend to adsorb more strongly to sediments than the less chlorinated or bromated congeners and can therefore remain stationary for long periods of time [35]. These compounds are released into the water by resuspension activities and can enter aquatic organisms through the gills, absorbing suspended solids, or in food, through which they can exert effects on fish life cycles [36].

### 4.3. PCBs and PBDEs in Water

In the water samples, the concentrations of PCBs and PBDEs reached 2.98 and 0.32 ng/mL respectively. In this study, the PCB and PBDE concentrations found in water were lower than those found in sediments. This may be partially explained by the volatility and particularly hydrophobic nature of these compounds [7]. It has been widely described that surface runoff, wastewater discharge, and untreated industrial and residential waste are the main contributors to the input of these pollutants to water bodies [37]. In particular, Lake Chapala, having light maritime activity, could claim that the local contribution of PCBs and PBDEs is minimal and, therefore, the ingress and distribution of their homologues may be the result of a combination of atmospheric sources [6] and the presence of point sources of pollution from the Lerma River.

Lake Chapala is a relatively shallow body of water that is exposed to high levels of UV light, especially around midday, and during the summer, which could accelerate the degradation of the most chlorinated or brominated PCBs and PBDEs by UV radiation [38]. Both PCBs and PBDEs can be hydroxylated (OH-PCB, OH-PBDE) by UV exposure as well as a variety of other chemical mechanisms, including metabolic transformation by living organisms and abiotic reactions with hydroxy radicals [39]. It is important to note that, in recent years, OH-PCBs and OH-PBDEs were detected in different environmental matrices, including water, sediments, and wildlife. This is a cause for particular concern, given that these hydroxylated compounds have a variety of toxic effects at smaller doses than the original PCBs and PBDEs, and they are known to be important endocrine disruptors [40].

### 4.4. Temporal Changes in PCB and PBDE Levels

In general terms, the pattern of PCB and PBDE homologues differed between 2018 and 2019, showing increased concentrations in 2019, especially in fish. There was also a tendency for PCB concentrations to increase from one year to the next in sediments and in water, which could be attributed to industrial agricultural, and urban development in the region, as well as to the proximity to the lake of the metropolitan area of Guadalajara and discharge from the Lerma River. The toxicity and persistence of these compounds make them a potential ecological risk to biota and to the human population near Lake Chapala. PCB concentrations also increased in sediments from 2018 to 2019. This reaffirms the hypothesis that sediments generate conditions that favor the persistence of this kind of highly hydrophobic compound. At the same time, the concentrations of PCBs in water and in the fish monitored in this study decreased from 2018 to 2019. It seems possible that these results of decrease in PBDEs are due to changes in the physicochemical conditions in the lake and the specific chemical characteristics of each analyte. In this study, the evidence points toward outflow of PBDEs from the water body (degradation and evaporation), considering the metabolic processes of the fish (excretion and reproduction).

### 4.5. Bioaccumulation of PCBs and PBDEs

Exposure to PCBs and PBDEs can harm fish in several ways related to their metabolic activity, genotoxicity, and reproduction. These contaminants may bioaccumulate in fish via two main mechanisms absorption of the compounds dissolved in the water through the skin and gills, or consumption through the diet [41]. Thus, determining the PCB and PBDE concentrations in both the sediments and water of Lake Chapala is key to determining their main mode of bioaccumulation in aquatic organisms.

In general terms, the three fish species had low values SBF and WBF. The congeners of low and intermediate molecular weights (tri, tetra, and pentad) had the highest values, which is likely because the least chlorinated or bromated compounds transfer more easily to aquatic organisms, mainly because they can more easily penetrate cell membranes [42]. However, in this study, we also found the highest levels of PCBs and PBDEs in the fish, which can be attributed to the fact that, in general, the larger homologues have lower degradation rates than those with lighter chlorination of bromination and, therefore, are maintained to a higher extent in aquatic systems, where they also tend to bioaccumulate [43]. This also depends on the diet preferences of each species, increasing with each level of the trophic web. Furthermore, it is known that low molecular weight congeners degrade more easily and do not bioaccumulate as much as heavier ones because their Kow (partition coefficient) is lower, indicating little lipophilicity. Those of high molecular weight tend to accumulate in lipids and, therefore, along the trophic chain. Physicochemical factors can also influence SBF and WBF in fish. Among these factors, the most influential include salinity, temperature, organic matter, and dissolved oxygen. Lake Chapala is considered slightly saline, so salinity is probably one of the most influential factors for WBF. It has been widely documented that the absorption of compounds through the gills is affected by salinity and changes in different ions [44].

In this study, we observed that *C. carpio* could potentially bioaccumulate a higher amount of these compounds compared to the other two species. It is also important to consider the trophic niche that each species occupies. The position in the food web of this species has important consequences on the levels of bioaccumulation of these kinds of contaminants. The highest levels of PCBs and PBDEs accumulation have been reported in carnivorous species, which feed mainly on smaller fish through the food chain [45]. There is evidence that *C. carpio* normally feeds on aquatic plants and supplements its diet with arthropods, zooplankton, and small fish, which makes this species omnivorous. Another characteristic to consider is that *C. carpio* prefers to occupy the bottom of water bodies, making this species susceptible to more contact with sediments. The results of this study are consistent and indicate that PCB and PBDE bioaccumulation in *C. carpio* in Lake Chapala was due mostly to superficial sediments [21,46]. The levels of PCBs and PBDEs found in *Cyprinus carpio* in this study are lower than those reported for the same species in Lake Erie (Canada and the United States) [47] and similar to those reported Büyük Menderes River in Turkey [48] and the shallow Dongting Lake in China [49].

In *O. aureus*, we also found evidence of bioaccumulation. Although this species is apparently invasive in the lake, the region has a strong economic interest in this species, and populations of *O. aureus* are estimated to be the most abundant in Lake Chapala, according to the records of local fishermen. This species is characterized by its ability to adapt to extreme conditions, such as low oxygen level and temperature changes. Although its diet is generally characterized as herbivorous, adults may also consume other small organisms. They tend to remain at the surface; however, they also sometimes frequent the bottom of water bodies. The levels of PCBs and PBDEs in this study are consistent with levels in *O. aureus* in Lake Victoria and Lake Tanganyika in Tanzania [50]. However, they are higher than those reported for *Oreochromis niloticus* in Burullus Lake in Egypt [51] and lower than those reported in *Tilapia mossambicus* of the Tibetan plateau [52].

*Chirostoma* spp. was the smallest species in the study, and generally had lower levels of bioaccumulation. This species can be considered carnivorous in that it feeds mainly on zooplankton. This species is important because some authors present it as a characteristic species of Lake Chapala. Pollution, eutrophication, and siltation in Lake Chapala in combination with fishery overexploitation in recent years have led to a marked reduction in the population of *Chirostoma spp*. However, this species can be considered an excellent environmental bioindicator for the lake. In this study, we found evidence of bioaccumulation in all three fish species. Although the levels of bioaccumulation of PCBs and PBDEs in the fish in this study were relatively low, they pose a risk to other wildlife that prey upon them, in which these compounds may further bioaccumulate and biomagnify, and which may suffer chronic effects on development, reproductive failure, liver damage, cancer, emaciation syndrome, and death [53,54,55].

## 5. Conclusions

The characteristics of Lake Chapala, such as fresh water, altitude, high evaporation, habitat of endemic species, and geographical location make it an early warning region for the monitoring of PCBs and PBDEs; it has also become a convenient region for the global scientific research related to POPs monitoring.

Different patterns of PCB and PBDE congeners in sediments, water, and fish species were identified and described in this study. The fish species with benthic habits had higher levels of bioaccumulation than the species with pelagic habits. *C. Carpio* presented the highest degree of bioaccumulation of PCBs and PBDEs. The levels of accumulation of PCBs and PBDEs increased from 2018 to 2019 in all three species.

Regarding the origin of the pollution levels in Lake Chapala, the distribution of the different congeners and homologues of PCBs and PBDEs in the water and sediments suggest that a part of these low and medium molecular weight chemical substances are transported from distant sources, such as through strong winds from the Guadalajara metropolitan area. However, a significant portion of these chemicals could be entering from the Lerma River, which is the longest inland river of Mexico, and through major industrial areas.

According to international reference points and the evidence found in this study on the levels of contamination of PCBs and PBDEs in water, sediments, and fish, it is probable that the concentrations of these chemical substances cause biological effects; therefore, it is a potential ecological risk. The levels of PCBs and PBDEs were low and do not currently represent a threat to the fish species themselves. However, they may represent a risk to the Lake Chapala wildlife and the human population.

Finally, although at this time the bioaccumulation of PCBs and PBDEs in the fish of Lake Chapala is low, this issue should not be ignored by the human population, and should continue to be considered in further studies.

## Figures and Tables

**Figure 1 toxics-09-00241-f001:**
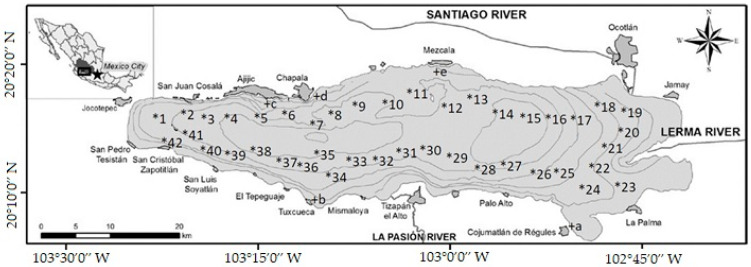
Map of Lake Chapala, adapted from Membrillo-Abad, Torres-Vera et al. 2016 [16], distribution of sediment, water, and fish samples. * With numbers; indicates sampling stations for water and sediments. + With letters; indicates fish sampling stations, also corresponds to landing of local fishermen (Cojumatlan de Regules, Tuxcueca, Ajijic, Chapala, and Mezcala).

**Figure 2 toxics-09-00241-f002:**
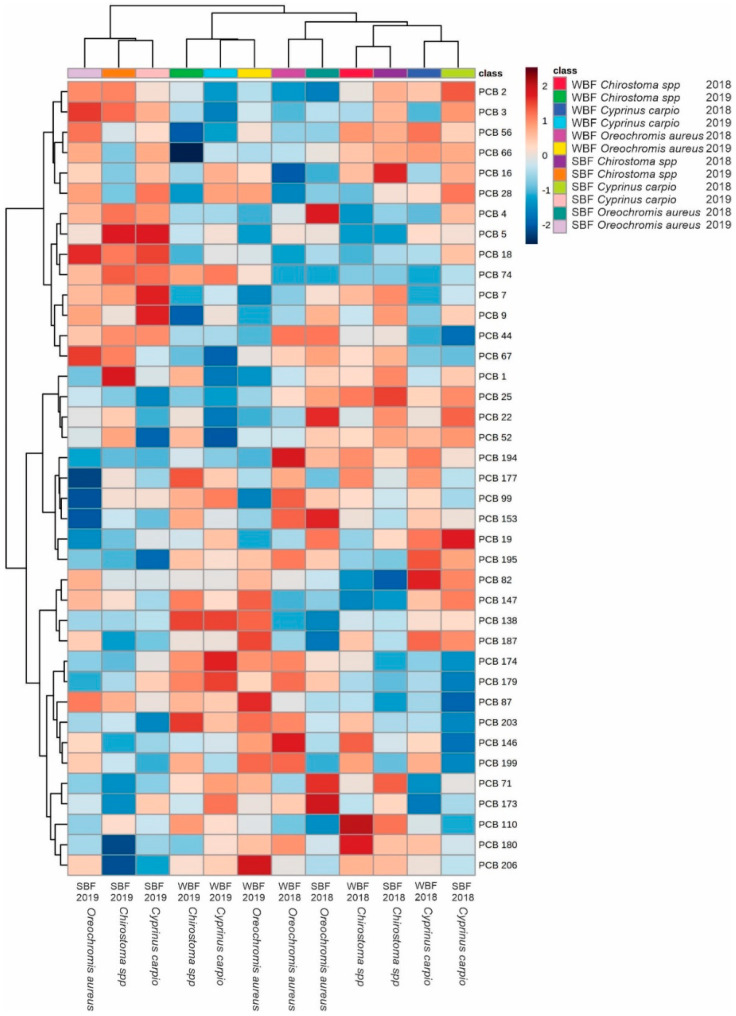
Heat map graph with hierarchical groupings for bioconcentration factors of PCBs in fish from Lake Chapala. In the upper part groupings with respect to the samples. On the left side groupings between analytes. FBS: bioaccumulation factor with respect to sediments. FBA: bioaccumulation factor with respect to water. The color scale represents the normalized bioaccumulation values depending on the samples; an intense red color represents a high value, while an intense blue color represents a low level. Distance measure: Euclidean. Clustering algorithm: ward. *t*-test/ANOVA.

**Figure 3 toxics-09-00241-f003:**
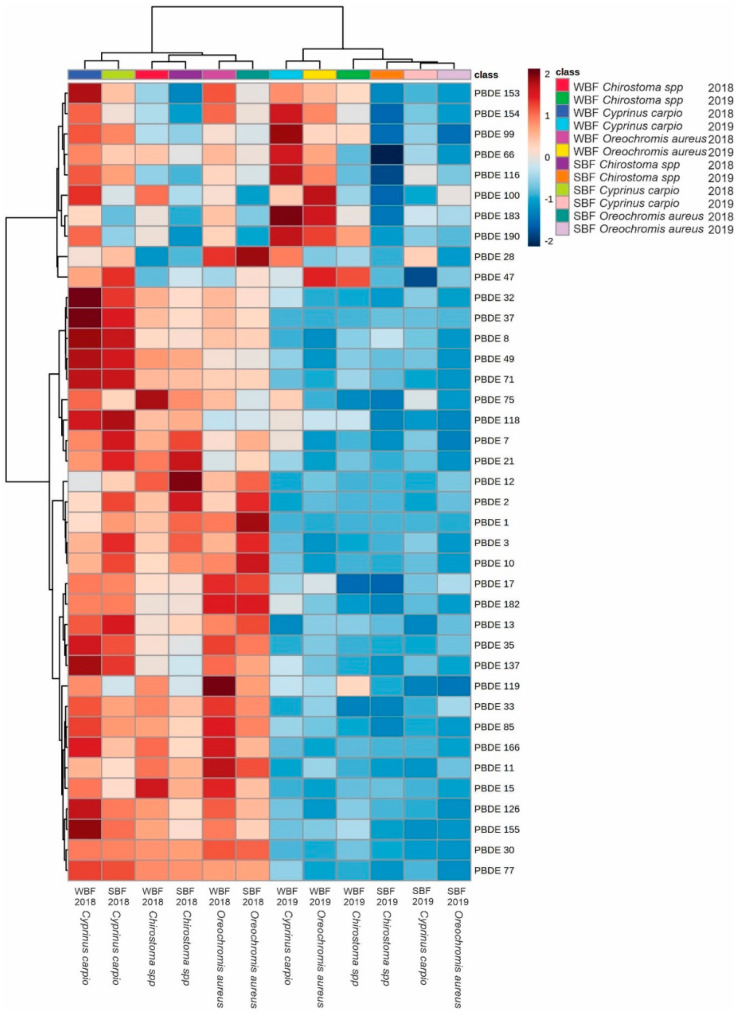
Heat map graph with hierarchical groupings for bioconcentration factors of PBDEs in fish from Lake Chapala. In the upper part groupings with respect to the samples. On the left side groupings between analytes. FBS: bioaccumulation factor with respect to sediments. FBA: bioaccumulation factor with respect to water. The color scale represents the normalized bioaccumulation values depending on the samples; an intense red color represents a high value, while an intense blue color represents a low level. Distance measure: Euclidean. Clustering algorithm: ward. *t*-test/ANOVA.

**Table 1 toxics-09-00241-t001:** Physicochemical parameters of sediments, water, and fish in October 2018 and May 2019.

Sediment		Electrical Conductivity (ms)	pH	Total Nitrogen (%)	Inorganic Carbon (%)	Organic Carbon (%)	Organic Matter (%)
	2018	1.01 ± 0.14 ^a^	7.45 ± 0.207 ^a^	0.22 ± 0.056 ^a^	0.39 ± 0.023 ^a^	1.85 ± 0.26 ^a^	3.2 ± 0.448 ^a^
	2019	1.22 ± 0.09 ^a^	7.30 ± 0.166 ^a^	0.15 ± 0.042 ^a^	0.56 ± 0.181 ^a^	2.04 ± 0.30 ^a^	3.54 ± 0.539 ^a^
Water		Temperature (°C)	Dissolved oxygen (mg/mL)	Electrical conductivity (uS/cm)	Total soluble solids (mg/L)	Salinity (ng/L)	Nitrates (mg/L)
	2018	23.54 ± 0.37 ^a^	5.12 ± 0.50 ^a^	767.88 ± 130.82 ^a^	519.36 ± 86.74 ^a^	0.000388 ± 0.0 ^b^	6.5743 ± 1.58 ^b^
	2019	23.57 ± 1.40 ^a^	5.26 ± 1.2 ^a^	869.84 ± 28.08 ^a^	581.32 ± 6.01 ^a^	0.000438 ± 0.0 ^a^	13.455 ± 4.89 ^a^
Fish		Fat (%)	Protein (%)	Weight (gr)	Length (cm)	Width (cm)	Length tail (cm)
*Chirostoma* spp.		4.17 ± 0.69 ^d^	72.19 ± 1.14 ^a^	7.00 ± 3.00 ^d^	5.00 ± 1.00 ^e^	0.70 ± 0.30 ^c^	2.0 ± 0.60 ^c^
*Cyprinus carpio*	2018	21.97 ± 3.68 ^b^	62.59 ± 1.90 ^b^	3339.00 ± 130 ^b^	28.00 ± 3.00 ^b^	9.20 ± 1.20 ^a^	6.20 ± 1.90 ^a^
*Oreochromis aureus*		14.71 ± 4.63 ^c^	58.54 ± 2.35 ^c^	213.00 ± 35 ^c^	15.50 ± 2.30 ^c^	5.10 ± 1.20 ^b^	3.70 ± 0.75 ^b^
*Chirostoma* spp.		3.71 ± 0.61 ^d^	70.38 ± 0.81 ^a^	7.50 ± 2.30 ^d^	7.00 ± 1.30 ^d^	0.70 ± 0.20 ^c^	1.80 ± 0.40 ^c^
*Cyprinus carpio*	2019	26.10 ± 2.33 ^a^	71.83 ± 3.35 ^a^	3542.00 ± 115.00 ^a^	32.00 ± 2.00 ^a^	9.50 ± 1.35 ^a^	6.80 ± 2.20 ^a^
*Oreochromis aureus*		12.30 ± 2.23 ^c^	59.92 ± 2.54 ^c^	205.00 ± 32.00 ^c^	17.00 ± 3.00 ^c^	5.80 ± 1.40 ^b^	4.10 ± 0.87 ^b^

Values show mean ± standard deviation of the different physicochemical parameters. Different letters in superscript indicate significant differences between rows, based on an ANOVA with Tukey test (≤0.05). Sediments *n* = 84; water = 84; *Chirostoma* spp. *n* = 40, *C. carpio n* = 40, and *O. aureus n* = 40.

**Table 2 toxics-09-00241-t002:** PCBs concentrations in matrices of: sediments, water, and fish species from Lake Chapala.

	2018					2019				
Name	Sediments	Water	*Chirostoma* spp.	*Cyprinus carpio*	*Oreochromis aureus*	Sediments	Water	*Chirostoma* spp.	*Cyprinus carpio*	*Oreochromis aureus*
PCB 1	2.31 ± 1.25	2.50 ± 1.24	1.28 ± 0.63	1.54 ± 0.72	1.43 ± 0.78	2.33 ± 1.54	2.14 ± 1.0	4.57 ± 2.05	3.32 ± 1.51	3.27 ± 2.84
PCB 2	2.45 ± 1.06	2.71 ± 1.49	1.32 ± 0.76	1.86 ± 0.90	1.04 ± 0.67	2.24 ± 1.26	2.21 ± 1.41	4.00 ± 1.76	3.45 ± 1.97	4.34 ± 2.57
PCB 3	2.07 ± 1.16	2.98 ± 1.75	1.11 ± 0.65	1.47 ± 0.77	1.07 ± 0.60	2.02 ± 1.41	2.61 ± 1.53	3.75 ± 0.17	3.55 ± 2.33	4.81 ± 1.70
PCB 4	2.15 ± 1.34	2.72 ± 1.71	0.91 ± 0.60	1.59 ± 0.84	1.78 ± 0.70	2.16 ± 1.31	2.45 ± 1.45	4.39 ± 2.60	4.72 ± 2.52	4.45 ± 1.85
PCB 5	2.30 ± 1.01	2.13 ± 1.73	0.30 ± 0.18	1.40 ± 0.90	1.27 ± 1.13	1.66 ± 0.78	2.51 ± 1.38	3.78 ± 1.80	4.89 ± 1.42	2.46 ± 0.06
PCB 7	2.47 ± 1.29	2.59 ± 1.49	1.30 ± 0.61	1.39 ± 1.02	1.35 ± 0.87	2.30 ± 1.55	2.48 ± 1.41	3.45 ± 2.78	4.34 ± 2.28	3.74 ± 2.21
PCB 9	2.28 ± 1.56	2.60 ± 1.68	1.30 ± 0.66	1.55 ± 1.09	1.50 ± 0.85	2.22 ± 1.39	2.48 ± 1.42	3.34 ± 2.33	5.52 ± 2.74	4.55 ± 2.60
PCB 16	2.14 ± 1.31	2.28 ± 1.45	1.79 ± 0.60	1.96 ± 1.16	1.36 ± 0.77	2.30 ± 1.66	1.73 ± 1.26	3.96 ± 3.18	5.17 ± 3.37	5.20 ± 3.78
PCB 18	2.51 ± 1.15	2.75 ± 1.86	1.23 ± 0.87	1.82 ± 1.12	1.51 ± 0.84	2.30 ± 1.30	2.49 ± 1.25	4.39 ± 2.99	5.54 ± 3.10	5.87 ± 2.10
PCB 19	1.98 ± 1.26	2.09 ± 1.48	1.24 ± 0.85	2.08 ± 0.90	1.58 ± 1.00	2.69 ± 1.61	1.85 ± 1.12	4.30 ± 2.14	5.29 ± 2.79	4.36 ± 3.18
PCB 22	2.14 ± 1.36	2.33 ± 1.32	1.40 ± 0.77	1.95 ± 1.09	1.70 ± 0.87	2.62 ± 1.42	2.13 ± 1.49	4.79 ± 3.25	4.14 ± 2.68	4.97 ± 2.72
PCB 25	2.35 ± 1.57	2.22 ± 1.51	1.69 ± 0.71	1.96 ± 1.11	1.74 ± 0.84	2.46 ± 1.57	1.87 ± 1.37	3.71 ± 2.95	3.67 ± 2.21	4.53 ± 2.67
PCB 28	2.05 ± 1.34	2.30 ± 1.44	1.10 ± 0.53	1.85 ± 0.87	1.21 ± 0.61	2.70 ± 1.57	2.20 ± 1.46	3.68 ± 3.30	6.07 ± 3.35	6.00 ± 2.93
PCB 44	2.48 ± 1.53	2.25 ± 1.39	1.39 ± 0.73	1.53 ± 1.00	1.87 ± 0.85	2.54 ± 1.56	2.27 ± 1.17	4.90 ± 3.13	5.64 ± 3.61	5.48 ± 2.94
PCB 52	2.35 ± 1.43	2.32 ± 1.48	1.71 ± 1.05	2.23 ± 0.89	1.76 ± 0.98	2.51 ± 1.40	1.94 ± 1.17	5.21 ± 2.95	3.15 ± 1.78	5.00 ± 4.07
PCB 56	2.44 ± 1.50	2.07 ± 1.36	1.60 ± 0.87	2.02 ± 0.98	1.57 ± 0.79	2.37 ± 1.29	2.22 ± 1.22	4.29 ± 3.79	5.09 ± 3.67	5.95 ± 3.82
PCB 66	2.34 ± 1.67	2.32 ± 1.27	1.84 ± 0.62	2.40 ± 1.21	1.71 ± 1.10	2.07 ± 1.56	2.36 ± 1.10	3.05 ± 2.19	5.97 ± 3.27	5.76 ± 3.03
PCB 67	2.80 ± 1.43	2.62 ± 1.25	1.84 ± 0.80	1.77 ± 1.01	2.13 ± 0.85	2.09 ± 1.43	2.28 ± 1.26	4.50 ± 3.17	3.72 ± 1.55	5.93 ± 3.89
PCB 71	2.37 ± 1.45	2.65 ± 1.75	1.75 ± 0.78	1.73 ± 0.98	1.96 ± 0.59	3.16 ± 1.29	1.83 ± 1.12	4.31 ± 3.32	5.43 ± 2.42	5.57 ± 2.78
PCB 74	2.30 ± 1.83	2.52 ± 1.11	0.45 ± 0.24	1.06 ± 0.81	0.55 ± 0.34	2.20 ± 1.29	1.91 ± 1.33	4.93 ± 3.02	5.92 ± 2.88	4.28 ± 3.20
PCB 82	2.74 ± 1.53	2.47 ± 1.51	1.38 ± 0.90	2.52 ± 0.92	2.00 ± 1.04	2.59 ± 1.47	2.01 ± 1.22	4.81 ± 3.65	5.52 ± 3.82	6.34 ± 2.44
PCB 87	3.10 ± 1.64	2.53 ± 1.73	1.54 ± 0.73	1.79 ± 1.18	1.99 ± 1.05	2.66 ± 1.43	1.98 ± 1.05	5.31 ± 2.91	5.62 ± 4.13	7.20 ± 1.60
PCB 99	2.49 ± 1.47	1.89 ± 1.23	1.41 ± 0.88	1.74 ± 1.24	1.89 ± 0.93	2.57 ± 1.61	1.69 ± 1.15	4.83 ± 3.57	5.56 ± 2.24	3.69 ± 2.89
PCB 110	2.84 ± 1.41	2.32 ± 1.34	1.88 ± 1.06	1.78 ± 1.22	1.58 ± 0.76	2.82 ± 1.48	1.95 ± 1.20	5.09 ± 2.72	5.02 ± 3.60	4.93 ± 3.76
PCB 138	2.80 ± 1.78	2.41 ± 1.56	1.54 ± 0.69	2.14 ± 1.01	1.70 ± 0.96	2.43 ± 1.75	1.43 ± 1.01	4.16 ± 2.67	4.52 ± 2.84	4.58 ± 2.76
PCB 146	2.61 ± 1.60	1.89 ± 1.46	1.51 ± 0.93	1.72 ± 1.04	1.87 ± 0.89	2.66 ± 1.68	1.87 ± 1.10	4.35 ± 2.69	5.10 ± 2.86	5.92 ± 3.17
PCB 147	2.48 ± 1.76	2.48 ± 1.58	1.16 ± 0.87	2.21 ± 0.98	1.51 ± 0.93	2.67 ± 1.91	1.77 ± 1.12	4.70 ± 3.12	4.54 ± 3.20	5.60 ± 2.67
PCB 153	2.69 ± 1.78	2.49 ± 1.45	1.31 ± 0.84	1.75 ± 1.09	2.00 ± 0.65	3.20 ± 1.77	2.04 ± 1.09	4.56 ± 3.23	4.08 ± 2.37	4.31 ± 3.92
PCB 173	2.23 ± 1.63	2.38 ± 1.33	1.16 ± 0.64	1.34 ± 1.02	1.64 ± 0.84	2.74 ± 1.41	1.81 ± 1.05	3.48 ± 2.57	4.59 ± 3.17	4.15 ± 2.36
PCB 174	2.72 ± 1.60	2.14 ± 1.48	1.14 ± 0.60	1.34 ± 0.85	1.69 ± 0.93	3.04 ± 1.83	1.71 ± 1.20	3.98 ± 3.05	5.06 ± 2.50	4.48 ± 2.20
PCB 177	2.69 ± 1.47	1.78 ± 1.31	1.40 ± 0.94	1.67 ± 1.10	1.46 ± 0.71	3.11 ± 1.64	1.64 ± 1.18	5.03 ± 2.81	4.29 ± 3.39	3.61 ± 2.67
PCB 179	2.70 ± 1.59	1.96 ± 1.10	0.85 ± 0.60	1.16 ± 0.76	1.77 ± 0.74	3.21 ± 1.64	1.51 ± 0.99	4.07 ± 2.71	5.72 ± 3.35	3.78 ± 2.69
PCB 180	2.35 ± 1.53	1.75 ± 1.32	1.42 ± 0.67	1.58 ± 0.78	1.47 ± 0.74	2.96 ± 1.79	1.81 ± 1.15	3.46 ± 2.14	4.63 ± 2.46	4.91 ± 2.73
PCB 187	2.65 ± 1.80	2.08 ± 1.23	1.16 ± 0.80	1.81 ± 0.78	1.16 ± 0.58	2.91 ± 1.83	1.90 ± 0.91	3.66 ± 2.71	3.71 ± 2.60	4.98 ± 2.81
PCB 194	2.58 ± 1.30	1.91 ± 1.20	1.39 ± 0.66	1.74 ± 0.82	1.72 ± 0.78	2.95 ± 1.62	2.03 ± 1.11	4.01 ± 2.55	3.96 ± 2.08	4.27 ± 2.44
PCB 195	2.39 ± 1.91	1.84 ± 1.16	0.79 ± 0.67	1.80 ± 0.92	1.52 ± 0.80	3.2 ± 1.71	1.78 ± 1.20	3.93 ± 2.88	3.70 ± 2.62	4.46 ± 2.90
PCB 199	2.76 ± 1.44	1.83 ± 1.15	1.08 ± 0.84	1.26 ± 0.88	1.41 ± 0.76	2.97 ± 1.46	1.95 ± 1.11	4.00 ± 2.38	3.68 ± 1.99	5.05 ± 2.69
PCB 203	2.73 ± 1.83	1.83 ± 1.22	1.26 ± 0.74	1.18 ± 0.72	1.64 ± 0.69	3.27 ± 1.88	1.47 ± 1.08	4.69 ± 3.34	3.93 ± 2.44	4.80 ± 2.66
PCB 206	2.52 ± 1.84	2.27 ± 0.85	1.38 ± 0.66	1.50 ± 0.93	1.38 ± 0.87	3.17 ± 1.84	1.59 ± 1.02	3.15 ± 2.31	3.81 ± 2.26	5.60 ± 3.34

Mean values ± standard deviation are presented. Units: concentration in ng/g in dry weight for samples of sediment (*n* = 84), *Chirostoma* spp. (*n* = 40), *C. carpio*, (*n* = 40), and *O. aureus* (*n* = 40). Units: ng/mL for water samples (*n* = 84). Three technical replicas were analyzed for each sample.

**Table 3 toxics-09-00241-t003:** PBDE concentrations in matrices of: sediments, water, and fish species from Lake Chapala.

	2018					2019				
Name	Sediments	Water	*Chirostoma* spp.	*Cyprinus carpio*	*Oreochromis aureus*	Sediments	Water	*Chirostoma* spp.	*Cyprinus carpio*	*Oreochromis aureus*
PBDE 1	0.18 ± 0.10	0.26 ± 0.16	0.14 ± 0.06	0.12 ± 0.11	0.20 ± 0.04	0.24 ± 0.14	0.25 ± 0.13	0.02 ± 0.04	0.02 ± 0.07	0.01 ± 0.03
PBDE 2	0.17 ± 0.10	0.29 ± 0.16	0.15 ± 0.07	0.13 ± 0.07	0.14 ± 0.02	0.23 ± 0.11	0.24 ± 0.14	0.02 ± 0.05	0.01 ± 0.02	0.03 ± 0.06
PBDE 3	0.22 ± 0.15	0.32 ± 0.16	0.15 ± 0.13	0.17 ± 0.06	0.17 ± 0.07	0.19 ± 0.11	0.26 ± 0.13	0.02 ± 0.04	0.03 ± 0.09	0.01 ± 0.03
PBDE 7	0.22 ± 0.11	0.28 ± 0.20	0.14 ± 0.10	0.15 ± 0.04	0.11 ± 0.05	0.29 ± 0.14	0.17 ± 0.12	0.03 ± 0.06	0.06 ± 0.09	0.02 ± 0.07
PBDE 8	0.26 ± 0.16	0.25 ± 0.18	0.13 ± 0.10	0.25 ± 0.04	0.14 ± 0.05	0.19 ± 0.13	0.26 ± 0.13	0.07 ± 0.09	0.04 ± 0.09	0.02 ± 0.07
PBDE 10	0.18 ± 0.12	0.24 ± 0.14	0.13 ± 0.05	0.16 ± 0.10	0.18 ± 0.06	0.24 ± 0.12	0.22 ± 0.13	0.04 ± 0.07	0.05 ± 0.10	0.03 ± 0.05
PBDE 11	0.26 ± 0.12	0.22 ± 0.15	0.14 ± 0.06	0.12 ± 0.08	0.18 ± 0.11	0.23 ± 0.13	0.19 ± 0.12	0.04 ± 0.07	0.03 ± 0.07	0.06 ± 0.10
PBDE 12	0.19 ± 0.13	0.26 ± 0.13	0.23 ± 0.05	0.11 ± 0.08	0.16 ± 0.08	0.24 ± 0.15	0.26 ± 0.14	0.03 ± 0.07	0.02 ± 0.06	0.05 ± 0.08
PBDE 13	0.25 ± 0.13	0.29 ± 0.16	0.14 ± 0.09	0.24 ± 0.03	0.22 ± 0.12	0.24 ± 0.13	0.18 ± 0.12	0.04 ± 0.09	0.02 ± 0.02	0.04 ± 0.08
PBDE 15	0.29 ± 0.12	0.18 ± 0.13	0.21 ± 0.06	0.16 ± 0.12	0.19 ± 0.05	0.18 ± 0.13	0.19 ± 0.09	0.03 ± 0.07	0.02 ± 0.08	0.01 ± 0.03
PBDE 17	0.24 ± 0.15	0.23 ± 0.13	0.14 ± 0.06	0.19 ± 0.11	0.21 ± 0.10	0.22 ± 0.18	0.19 ± 0.15	0.03 ± 0.07	0.08 ± 0.11	0.10 ± 0.13
PBDE 21	0.17 ± 0.11	0.21 ± 0.13	0.19 ± 0.06	0.18 ± 0.04	0.11 ± 0.09	0.22 ± 0.14	0.17 ± 0.13	0.05 ± 0.09	0.06 ± 0.11	0.03 ± 0.07
PBDE 28	0.20 ± 0.14	0.22 ± 0.16	0.10 ± 0.09	0.14 ± 0.08	0.17 ± 0.15	0.26 ± 0.16	0.23 ± 0.14	0.12 ± 0.09	0.17 ± 0.12	0.12 ± 0.10
PBDE 30	0.26 ± 0.14	0.25 ± 0.16	0.15 ± 0.06	0.16 ± 0.10	0.17 ± 0.10	0.21 ± 0.15	0.16 ± 0.11	0.04 ± 0.08	0.04 ± 0.09	0.03 ± 0.08
PBDE 32	0.24 ± 0.12	0.19 ± 0.16	0.10 ± 0.06	0.18 ± 0.02	0.10 ± 0.06	0.28 ± 0.18	0.19 ± 0.14	0.02 ± 0.06	0.06 ± 0.11	0.02 ± 0.05
PBDE 33	0.26 ± 0.13	0.21 ± 0.14	0.12 ± 0.06	0.14 ± 0.09	0.14 ± 0.13	0.20 ± 0.16	0.22 ± 0.07	0.01 ± 0.03	0.03 ± 0.07	0.05 ± 0.10
PBDE 35	0.24 ± 0.18	0.20 ± 0.11	0.11 ± 0.05	0.23 ± 0.13	0.21 ± 0.10	0.25 ± 0.14	0.21 ± 0.12	0.03 ± 0.06	0.03 ± 0.08	0.05 ± 0.11
PBDE 37	0.27 ± 0.17	0.23 ± 0.13	0.16 ± 0.05	0.31 ± 0.07	0.17 ± 0.10	0.18 ± 0.14	0.24 ± 0.12	0.04 ± 0.09	0.04 ± 0.08	0.03 ± 0.06
PBDE 47	0.24 ± 0.14	0.26 ± 0.16	0.17 ± 0.08	0.22 ± 0.09	0.18 ± 0.07	0.24 ± 0.17	0.18 ± 0.12	0.16 ± 0.09	0.13 ± 0.11	0.16 ± 0.11
PBDE 49	0.24 ± 0.16	0.23 ± 0.15	0.16 ± 0.07	0.25 ± 0.10	0.11 ± 0.10	0.23 ± 0.15	0.17 ± 0.11	0.04 ± 0.08	0.04 ± 0.09	0.01 ± 0.02
PBDE 66	0.26 ± 0.13	0.24 ± 0.14	0.18 ± 0.09	0.20 ± 0.10	0.19 ± 0.10	0.27 ± 0.12	0.19 ± 0.12	0.11 ± 0.09	0.18 ± 0.12	0.15 ± 0.13
PBDE 71	0.21 ± 0.16	0.21 ± 0.13	0.18 ± 0.07	0.28 ± 0.09	0.16 ± 0.09	0.26 ± 0.15	0.18 ± 0.15	0.08 ± 0.11	0.06 ± 0.10	0.05 ± 0.08
PBDE 75	0.26 ± 0.14	0.19 ± 0.08	0.19 ± 0.06	0.15 ± 0.12	0.12 ± 0.10	0.24 ± 0.15	0.19 ± 0.11	0.03 ± 0.08	0.11 ± 0.13	0.05 ± 0.09
PBDE 77	0.24 ± 0.15	0.23 ± 0.16	0.18 ± 0.07	0.21 ± 0.11	0.17 ± 0.06	0.34 ± 0.23	0.25 ± 0.12	0.06 ± 0.10	0.09 ± 0.12	0.05 ± 0.11
PBDE 85	0.27 ± 0.17	0.22 ± 0.14	0.15 ± 0.05	0.18 ± 0.16	0.19 ± 0.07	0.31 ± 0.16	0.20 ± 0.12	0.04 ± 0.07	0.07 ± 0.11	0.06 ± 0.09
PBDE 99	0.26 ± 0.15	0.24 ± 0.14	0.16 ± 0.07	0.23 ± 0.09	0.18 ± 0.08	0.28 ± 0.18	0.16 ± 0.09	0.13 ± 0.09	0.18 ± 0.09	0.13 ± 0.08
PBDE 100	0.29 ± 0.15	0.21 ± 0.12	0.19 ± 0.07	0.20 ± 0.10	0.15 ± 0.08	0.24 ± 0.17	0.17 ± 0.11	0.11 ± 0.09	0.13 ± 0.11	0.18 ± 0.11
PBDE 116	0.26 ± 0.16	0.24 ± 0.14	0.14 ± 0.08	0.20 ± 0.11	0.17 ± 0.07	0.29 ± 0.17	0.21 ± 0.11	0.11 ± 0.08	0.19 ± 0.13	0.16 ± 0.09
PBDE 118	0.20 ± 0.13	0.21 ± 0.13	0.16 ± 0.06	0.21 ± 0.09	0.12 ± 0.09	0.28 ± 0.16	0.18 ± 0.12	0.11 ± 0.09	0.12 ± 0.11	0.11 ± 0.09
PBDE 119	0.26 ± 0.16	0.20 ± 0.14	0.17 ± 0.06	0.17 ± 0.11	0.21 ± 0.08	0.26 ± 0.19	0.18 ± 0.10	0.13 ± 0.08	0.11 ± 0.11	0.11 ± 0.08
PBDE 126	0.29 ± 0.16	0.22 ± 0.13	0.18 ± 0.05	0.27 ± 0.00	0.22 ± 0.08	0.29 ± 0.17	0.19 ± 0.10	0.06 ± 0.09	0.05 ± 0.09	0.02 ± 0.05
PBDE 137	0.26 ± 0.16	0.22 ± 0.13	0.10 ± 0.07	0.23 ± 0.07	0.17 ± 0.08	0.30 ± 0.17	0.17 ± 0.11	0.03 ± 0.07	0.07 ± 0.12	0.04 ± 0.08
PBDE 153	0.28 ± 0.16	0.20 ± 0.11	0.10 ± 0.06	0.18 ± 0.11	0.16 ± 0.07	0.27 ± 0.17	0.16 ± 0.11	0.10 ± 0.08	0.12 ± 0.10	0.11 ± 0.09
PBDE 154	0.28 ± 0.15	0.21 ± 0.11	0.12 ± 0.08	0.19 ± 0.10	0.18 ± 0.10	0.31 ± 0.20	0.16 ± 0.10	0.10 ± 0.07	0.16 ± 0.11	0.13 ± 0.09
PBDE 155	0.22 ± 0.19	0.16 ± 0.13	0.14 ± 0.07	0.21 ± 0.08	0.15 ± 0.06	0.35 ± 0.19	0.13 ± 0.09	0.05 ± 0.08	0.04 ± 0.09	0.04 ± 0.07
PBDE 166	0.31 ± 0.15	0.19 ± 0.15	0.12 ± 0.06	0.14 ± 0.09	0.15 ± 0.06	0.27 ± 0.17	0.23 ± 0.11	0.03 ± 0.06	0.03 ± 0.06	0.01 ± 0.06
PBDE 182	0.23 ± 0.13	0.23 ± 0.14	0.10 ± 0.05	0.15 ± 0.08	0.19 ± 0.07	0.33 ± 0.17	0.18 ± 0.12	0.02 ± 0.05	0.07 ± 0.10	0.04 ± 0.08
PBDE 183	0.29 ± 0.16	0.21 ± 0.14	0.12 ± 0.07	0.13 ± 0.09	0.14 ± 0.07	0.27 ± 0.18	0.16 ± 0.11	0.10 ± 0.08	0.15 ± 0.10	0.14 ± 0.08
PBDE 190	0.34 ± 0.17	0.20 ± 0.13	0.11 ± 0.07	0.15 ± 0.08	0.12 ± 0.00	0.34 ± 0.21	0.16 ± 0.09	0.11 ± 0.07	0.14 ± 0.11	0.13 ± 0.08

PBDE bioaccumulation values above 1.00 were more frequent among WBF than SBF. For 2018, the value calculated for WBF corresponding to PBDE 15 (1.16) was higher in *Chirostoma* spp., while for *C. carpio* PBDEs 8, 35, 37, 49, 71, 126, 137, and 155, with values of 1.02, 1.14, 1.36, 1.09, 1.34, 1.22, 1.04, and 1.32, respectively, were the most important. In the case of *O. aureus*, these were PBDE 15 (1.07), PBDE 35 (1.02), and PBDE 119 (1.06). In 2019, WBF values above 1.00 were not observed for *Chirostoma* spp., but for *C. carpio* and *O. aureus*, PBDE 99 (1.07) and PBDE 100 (1.05), respectively, were the most important analytes (Figure 3).

## Data Availability

The data presented in this study are available in this article.

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
