# Peer review of "Bioaccumulation of PCBs and PBDEs in Fish from a Tropical Lake Chapala, Mexico"

_toxics, 2021, doi:10.3390/toxics9100241_

Round 1
Reviewer 1 Report
Dear Authors,
The research article entitled ‘Bioaccumulation of PCBs and PBDEs in fish from lake Chapala, Mexico’ is written in a very interesting way. The topic of the article contains crucial issues related to the functioning and characteristics (contamination) of fish living in the Chapala Lake, Mexico in terms of potential risk for the rest of the biota and for the local people. From this point of view, this article should be published with the current contents, however, for being accepted, some major changes should be done (attached pdf).

Author Response
Dear reviewer,
Thank you for your time. Your comments and suggestions were well received. They certainly improved the work.
Point 1: My proposition of the title is “Bioaccumulation of PCBs and PBDEs in fish, water and sediments from the Chapala Lake, Mexico”
Response 1: Although the proposal presented is interesting, we consider that it would be more convenient not to add the water and sediment matrices to the title, since traditionally the bioaccumulation phenomenon takes place in living organisms and has been described in this way. We prefer to change the title to the following: Bioaccumulation of PCBs and PBDEs in fish from a tropical Lake-Lake Chapala, Mexico
Abstract
Point 2: L15: I suggest not to start the sentence, and in this case the entire article, with the phrase ‘We determined...’. This is not stylistically correct. Please start with the short description of the topic.
Response 2: Agree, the suggestion is well received, we correct the phrase "We determine" throughout the document, we also try to start each paragraph with a short description.
Point 3: L18: gas chromatography coupled with mass spectrometry.
Response 3: Agree, the sentence "gas chromatography coupled with mass spectrometry" was corrected.
Point 4: L24-27: Please rewrite the last two sentences.
Response 4: Agree, the last two sentences of the abstract were rewritten.
Introduction
Point 5: L32: If added the abbreviation POPs then it should be written precisely persistent organic pollutants instead of compounds.
Response 5: Agree, the expression persistent organic pollutants for the abbreviation of POPs was corrected.
Point 6: L33-35: The examples of the literature of the places where POPs have been detected (similar to the authors’ research) should be added after this statement (please see the review article https://doi.org/10.3390/w13131739 with the included articles).
Response 6: The suggested reference was very interesting, it was analyzed including the articles, some of them were used in the introduction.
Point 7: L47: ‘The Chapala Lake’. Please change throughout the manuscript.
Response 7: We believe that the expression Lake Chapala adequately describes the name of the study area, and that is how it has been identified in previous works.
Point 8: L79: Delete ‘in which’. Instead of ‘PCB’ and ‘PBDE’ please write ‘PCBs’ and ‘PBDEs’ to be consistent.
Response 8: Agree, the sentence was corrected, "in which" was removed and “PCBs” and “PBDEs” were corrected.
Point 9: L91-92: Why the authors chose different months for the research in 2018 and 2019? Was there any specific reason?
Response 9: The reason was that they tried to take representative samples in the months corresponding to the local seasonal fluctuation, corresponding to the dry season (May 2019) and the rainy season (October 2018). In addition, these months could correspond to the moments when there is higher and lower water level in the Lake Chapala in a seasonal cycle. This information was added in the document.
Materials and Methods
Point 10: L105: How the samples were collected (especially sediments and water, and from what depth)? It is missing in the text.
Response 10: Agree, a part of the materials and methods section was rewritten to be more precise and include more information related to sample collection.
Point 11: L112-124, 127: In my opinion, it is better to write this paragraph in 3rd person, just like the whole article.
Response 11: Agree, the entire document was reviewed, and it was written in the third person.
Point 12: L143: I think that the title of the paragraph ‘Gas chromatography and mass spectrometry (GC/MS)’ is unnecessary. In the next paragraph, the authors describe the determination of PCBs using GC-MS, so they describe the entire analysis in a way. I suggest combining both paragraphs and describe all the analysis in one paragraph (both PCBs and PBDEs). There are too many short paragraphs.
Response 12: Agree, sections were combined to reduce short paragraphs.
Point 13: L157-178: The authors have to decide whether the whole article is written in 1st or 3rd person. It must be consistent. Otherwise, the article is hard to read.
Response 13: Your comment is well received. The document was revised and rewritten in the third person.
Results
Point 14: L196-206: the results are described as in the report. A scientific publication is not a report. Even if results are presented, each paragraph must contain some introduction to the topic under discussion. Please rewrite.
Response 14: Agree, added a short and concise introduction in each paragraph.
Point 15: L207: Please check the significant figures in the Table 1. The table title is misleading. The authors did not measure given parameters from 2018 to 2019, but only in two months in 2018 and 2019.
Response 15: Agree, the significance of the data in Table 1 was revised. The title was also rewritten to make it more appropriate.
Point 16: L216: I suggest changing the tables to graphs that will show changes in concentration levels of determined compounds. Presenting such a large number of results in a table is illegible and adds nothing. The authors might consider including result tables in Supplementary Materials.
Response 16: We agree is that they are very extensive tables. We really tried to replace them with more condensed graphics, but a lot of important information was lost in the attempt. Our opinion is to keep these tables in the document as they are.
Point 17: L217-231: Too many repetitions of ‘On the other hand’.
Response 17: Agree, the repetitions of the expression "On the other hand" were corrected.
Point 18: L265: Please do not start the paragraph like a report.
Response 18: Agree, an introduction was included in each paragraph.
Point 19: L305: The same as in L216.
Response 19: As in the previous comment, we believe it is necessary to maintain the tables, this way allows us to see the results broadly, replace them with graphs, they would also have to be very extensive and there is a risk of losing valuable information.
Discussion
Point 20: L316: BCP? Again, starting like a report, not like a scientific paper.
Response 20: Agree, an introduction was included in each paragraph.
Conclusions
Point 21: L459-460: Please rewrite this sentence. Maybe like this would be better: ‘Different patterns of PCB and PBDE congeners in sediments, water and fish species were identified and described in this study’.
Response 21: Agree, the sentence was written “Different patterns of PCB and PBDE congeners in sediments, water and fish species were identified and described in this study”.
Point 22: My overall impression is that the Conclusion section is too short. It ends in such a way that the reader has the impression that there will be more conclusions further.
Response 22: Agree, conclusions were added.
Reviewer 2 Report
The authors monitored the concentrations of PCBs and PBDEs in sediment and water from Lake Chapala and determined levels of bioaccumulation of PCBs and PBDEs in Chirostoma spp, Cyprinus carpio and Oreochromis aureus. The results seem interesting. However, some modifications need to be made before publication in toxics.
- The keywords are not appropriate. You may substitute with “PCBs, PBDEs, fish, bioaccumulation, sediments, water”.
- Relevant references are needed to provide to support descriptions. For example, “and the past 30 years…have been detected” in lines 33-35. “although the production…and improvised dumps” in lines 38-42.
- 168 sediment and water samples were collected from 42 sampling stations, and 120 samples of three fish species (Chirostoma spp, Cyprinus carpio and Oreochromis aureus) from five local fishing stations were collected during October 2018 and May 2019. Are the samples representative in terms of location and season? Are there any parallel samples for each site? How does this correlate with the sample number in Table 1 (Sediments n = 42; Water =42; Fish n = 20) and Table 2 (Sediments n = 42; Fish n = 40; Water n=42)?
- How are the water samples pretreated for analysis? What is the recovery of the treatment method? What is the limit of detection and limit of quantification for PCBs and PBDEs?
- Section 2.6, Identification and quantification of PBDEs are not included. Perhaps the title should be changed.
- What is the difference of organic carbon and organic matter in Table 1?
- The title of section 4.1 is BCP and PBDE levels. Typo error: “BCP” should be replaced with “PCBs”. Moreover, does this mean the levels of BCP and PBDE in fish?
- The units are not in consistent format. For example, ppt and mg/L.
- The whole fish is used in the assessment of bioaccumulation. The authors may consider use different organs/tissues, especially those edible parts to evaluate the risk of fish consumption.
Author Response
Dear reviewer,
Thank you for your time. Your comments and suggestions were well received. They certainly improved the work.
Point 1: The keywords are not appropriate. You may substitute with “PCBs, PBDEs, fish, bioaccumulation, sediments, water”.
Response 1: It is a good proposal; however, we believe it is convenient that the keywords of the abstract are the same as those of the title, we believe that the keywords we have are adequate and increase the possibilities for our document to be found in the searches.
Point 2: Relevant references are needed to provide to support descriptions. For example, “and the past 30 years…have been detected” in lines 33-35. “although the production…and improvised dumps” in lines 38-42.
Response 2: Agree, bibliographic references were added to support the paragraphs of the introduction.
Point 3: 168 sediment and water samples were collected from 42 sampling stations, and 120 samples of three fish species (Chirostoma spp, Cyprinus carpio and Oreochromis aureus) from five local fishing stations were collected during October 2018 and May 2019. Are the samples representative in terms of location and season? Are there any parallel samples for each site? How does this correlate with the sample number in Table 1 (Sediments n = 42; Water =42; Fish n = 20) and Table 2 (Sediments n = 42; Fish n = 40; Water n=42)?
Response 3: Agree, part of materials and methods was rewritten, and data was added to clarify the information. We believe that the number of samples, the dates of collection and the distribution of the sampling sites are representative. We also consider that they are adequate because it allowed us to process them properly.
Point 4: How are the water samples pretreated for analysis? What is the recovery of the treatment method? What is the limit of detection and limit of quantification for PCBs and PBDEs?
Response 4: Agree, information on detection limits of the studied analytes and recoveries of PCBs and PBDEs was also added.
Point 5: Section 2.6, Identification and quantification of PBDEs are not included. Perhaps the title should be changed.
Response 5: Agree, some sections were combined and the title was corrected.
Point 6: What is the difference of organic carbon and organic matter in Table 1?
Response 6: Organic carbon is contained in the soil organic fraction, which consists of the cells of microorganisms, plant and animal residues at various stages of decomposition, stable "humus" synthesized from residues, and highly carbonized compounds such as charcoal, graphite and coal (elemental forms of Carbon). Organic matter has been defmed as the organic fraction of soil, including plant, animal, and microbial residues, fresh and at all stages of decomposition, and the relatively resistant soil humus. doi.org/10.2136/sssabookser5.3.c34
Point 7: The title of section 4.1 is BCP and PBDE levels. Typo error: “BCP” should be replaced with “PCBs”. Moreover, does this mean the levels of BCP and PBDE in fish?
Response 7: Agree, the title of section 4.1 was rewritten.
Point 8: The units are not in consistent format. For example, ppt and mg/L.
Response 8: Agree, units were reviewed throughout the document.
Point 9: The whole fish is used in the assessment of bioaccumulation. The authors may consider use different organs/tissues, especially those edible parts to evaluate the risk of fish consumption.
Response 9: In this work, only the whole fish was used. However, sectioning the fish is a good idea, especially now that we have the background of the whole fish, our working group will consider it for future work. Thanks for the recommendation.
Round 2
Reviewer 1 Report
Dear Authors,
Thank you for the manuscript corrections and your reply to all my comments. In my opinion, the article is improved and for sure it is worth publishing. Although, I still have some small comments to reconsider. Please find them below:
‘Bioaccumulation of PCBs and PBDEs in fish from a tropical Lake‐lake Chapala, Mexico’. The title is much better like this. The only thing I was wondering maybe it is better to write ‘Bioaccumulation of PCBs and PBDEs in fish from the tropical Lake Chapala, Mexico’ to avoid repeating the word ‘lake’.
L228: The ‘a’ in word ‘Chapala’ is missing.
L229: ‘between’ not ‘be-tween’
Reviewer 2 Report
The authors have improved the manuscript according to the comments, and the manuscript can be accepted in present form.